# SNN-RAT: Robustness-enhanced Spiking Neural Network through Regularized Adversarial Training

**Jianhao Ding**
School of Computer Science
Peking University
Beijing, China 100871
djh01998@stu.pku.edu.cn

**Tong Bu**
Institution for Artificial Intelligence
School of Computer Science
Peking University
Beijing, China 100871
putong30@pku.edu.cn

**Zhaofei Yu**[*]
Institute for Artificial Intelligence
School of Computer Science
Peking University
Beijing, China 100871
yuzf12@pku.edu.cn

**Tiejun Huang**
School of Computer Science
Peking University
Beijing, China 100871
tjhuang@pku.edu.cn

**Jian K. Liu**
School of Computing
University of Leeds
Leeds LS2 9JT
j.liu9@leeds.ac.uk

## Abstract

Spiking neural networks (SNNs) are promising to be widely deployed in real-time and safety-critical applications with the advance of neuromorphic computing. Recent work has demonstrated the insensitivity of SNNs to small random perturbations due to the discrete internal information representation. The variety of training algorithms and the involvement of the temporal dimension pose more threats to the robustness of SNNs than that of typical neural networks. We account for the vulnerability of SNNs by constructing adversaries based on different differentiable approximation techniques. By deriving a Lipschitz constant specifically for the spike representation, we first theoretically answer the question of how much adversarial invulnerability is retained in SNNs. Hence, to defend against the broad attack methods, we propose a regularized adversarial training scheme with low computational overheads. SNNs can benefit from the constraint of the perturbed spike distance's amplification and the generalization on multiple adversarial $\epsilon$-neighbourhoods. Our experiments on the image recognition benchmarks have proven that our training scheme can defend against powerful adversarial attacks crafted from strong differentiable approximations. To be specific, our approach makes the black-box attacks of the Projected Gradient Descent attack nearly ineffective. We believe that our work will facilitate the spread of SNNs for safety-critical applications and help understand the robustness of the human brain. The code is available at https://github.com/putshua/SNN-RAT.

## 1 Introduction

Spiking Neural Networks (SNNs), unlike traditional Analog Neural Networks (ANNs), mimic the neuronal behaviours of a biological brain through spatio-temporal dynamics and spike representation [Gerstner et al., 2014], which represent the forefront of neural networks [Maass, 1997, Zenke et al., 2021]. The neurons evolve their membrane potentials as time goes by and transmit discrete

---

[*]Corresponding author

36th Conference on Neural Information Processing Systems (NeurIPS 2022).

information by 0 (nothing) and 1 (a spike). After the transmission, the membrane potentials are reset to the rest value and wait for the incoming input. Currently, the training algorithm of SNNs is a hot research topic due to the unique discrete spike activation. This results in a fundamental difference in training methods compared with ANNs. The rise of neuromorphic computing has allowed SNNs to run with more compatible hardware and a lower energy cost [Pei et al., 2019, DeBole et al., 2019, Davies et al., 2018, Nieves and Goodman, 2021, Fang et al., 2020]. The combination of SNN and neuromorphic hardware can enable numerous applications, such as spatio-temporal pattern recognition and high-speed detection [Wu et al., 2018a, Xu et al., 2020, Kim et al., 2020, Kheradpisheh and Masquelier, 2020, Zenke and Neftci, 2021].

For safety-critical applications such as autonomous driving, the reliability of the system becomes crucial, especially the robustness of the model against perturbations, such as additive Gaussian noise. Among all the perturbations, the adversarial attack is one of the most powerful categories [Szegedy et al., 2014, Goodfellow et al., 2015]. It can generate subtle perturbations that are usually neglected by the human perception system. The perturbations, however, can deteriorate the capability of the system, that is, the model produces incorrect labels with a high probability. This could have serious impacts on those safety-related applications, where a single failure could have devastating results. Up to now, a wide variety of adversarial attack methods have been proposed [Madry et al., 2018]. The vulnerability holds even from a model that is trained for the same task but with a different architecture.

The operation mechanism and structure of SNNs are similar to those of the biological brain, and studying its response to perturbation can help us understand how the human brain works. SNNs are recognized as a new potential candidate with adversarial robustness due to its input coding and neuronal dynamics [Perez-Nieves et al., 2021, Leontev et al., 2021]. Among the coding schemes commonly used in SNN, constant input coding is considered to be more susceptible to disturbances than others, like Poisson coding [Sharmin et al., 2020]. Therefore, Kundu et al. [2021b] suggested that careful training is required for constant input coding. In this setting, SNNs are now facing more challenges than typical ANNs. Because the key to constructing SNNs gradient-based attacks is back-propagation, which is the same as that of ANNs. However, compared with ANNs, SNNs can learn through various gradient approximations. Therefore, combining various differentiable approximations and attack methods will pose a more severe threat to SNNs [Liang et al., 2021].

The inter-layer communication of SNNs is through spikes with a time dimension, which is very different from ANNs. Therefore, one question can be raised naturally: whether and to what extent does spike communication detain adversarial invulnerability? And, are there training tools that can help SNNs defend against the threats described above? This paper aims to extend the Lipschitz analysis theory to spike representation and propose a more robust training algorithm on this basis. Our main contributions are summarized as follows:

- We design and summarize different differentiable approximations which can be deployed in gradient-based attacks to show the vulnerability of SNNs. Backward pass through time and rate are found to be capable of constructing stronger attacks.

- We theoretically analyze the $l_2$ perturbation distance on the representation of spikes and give a mathematical expression of the spiking Lipschitz constant.

- We propose a regularized adversarial training scheme for SNNs. It not only constrains the spiking Lipschitz constant but also exploits the mixture of the identified strong single-step adversarial attacks.

- Our experiments show that the proposed scheme can significantly improve the adversarial robustness in the image recognition tasks. The trained model exhibits better resistance under stronger Projected Gradient Descent attacks compared to the vanilla models.

## 2 Background and Related Work

### 2.1 Robustness of Spiking Neural Networks

The concept of spike representation is where SNNs are different from ANNs. Typically, the input is encoded as sequences $(T \times N_0)$, with $T$ denoting the total number of time-steps and $N_0$ denoting the number of the input nodes. The neurons in SNNs, like ANNs, receive a linear combination of

the previous layer's output. The mechanism of leaky-integrate-fire (LIF) leads to the non-linearity nature of SNNs. Overall, the dynamic of the membrane potential $\boldsymbol{m}^l(t)$ of neurons in layer $l$ ($l = 1, 2, \cdots, L$) at time-step $t$ ($t = 1, 2, \cdots, T$) can be described by:

$$\boldsymbol{m}^l(t^-) = \boldsymbol{m}^l(t-1) + \boldsymbol{W}^l \boldsymbol{s}^{l-1}(t), \tag{1}$$

$$\boldsymbol{s}^l(t) = H(\boldsymbol{m}^l(t^-) - \theta), \tag{2}$$

$$\boldsymbol{m}^l(t) = \lambda \boldsymbol{m}^l(t^-)(1 - \boldsymbol{s}^l(t)), \tag{3}$$

where $\boldsymbol{s}^l(t)$ denotes the binary spikes of neurons in layer $l$ at time $t$, which equals to 1 if there is spike. $\boldsymbol{m}^l(t^-)$ denotes the instantaneous state of membrane potential before triggering a spike, which accumulates the weighted input from presynaptic neurons in the last layer. When the potential exceeds the predefined threshold $\theta$, a spike is generated at time $t$, and the membrane potential $\boldsymbol{m}^l(t)$ is reset to zero. Otherwise the membrane potential $\boldsymbol{m}^l(t)$ either leaks by $\lambda$ for LIF model ($\lambda \in (0, 1)$), or maintains for Integrate-and-Fire (IF) model ($\lambda = 1$).

The leaking potentials described above intuitively smooth the noise in the input current, which is an appealing characteristic for trustworthy neural networks. Indeed, SNNs can demonstrate robustness under certain conditions. Sharmin et al. [2020] presented their pioneering work on input discretization and leaky rate. They highlighted that specific input coding could improve the robustness of SNN. Up to now, Poisson coding, latency coding, and time-to-first-spike coding have been proven to have an effect on small datasets [Sharmin et al., 2020, Leontev et al., 2021, Nomura et al., 2022]. Sharmin et al. [2019] also found that SNNs trained by surrogate functions can boost the robustness of Poisson coding. Whereas, SNNs are not completely secure. Marchisio et al. [2020] pointed out that black-box attacks can also attack SNNs. El-Allami et al. [2021] searched for the structural parameters of SNNs to improve the robustness. Among rate coding, constant input coding (or direct coding) is shown to detain less robustness, which has been discussed in several papers [Kim et al., 2022, Kundu et al., 2021b]. Based on careful observations, Kundu et al. [2021b] asserted that fine training is essential for the robustness of direct-coding SNNs. Motivated by this, we propose a regularizer that can promote robust training schemes for SNNs.

## 2.2 Gradient-based Adversarial Attacks

Most adversarial attacks consider a perturbation $\boldsymbol{\delta}$ in an $l_p$ ball around clean data $\boldsymbol{x}$ and can be formulated as an optimization problem:

$$\arg\max_{\boldsymbol{\delta}} \mathcal{L}(f(\boldsymbol{x} + \boldsymbol{\delta}; \boldsymbol{W}), y) \quad s.t. \ \|\boldsymbol{\delta}\|_p \leq \epsilon, \tag{4}$$

where $y$ is the target, $\mathcal{L}$ is the loss function, $f$ denotes the network with $\boldsymbol{W}$ as parameters, and $\epsilon$ is the parameter that guarantees the perturbation is imperceptible. Here, we introduce two widely adopted gradient-based adversarial attack algorithms: Fast Gradient Sign Method (FGSM) and Projected Gradient Descent method (PGD).

**FGSM**. As one of the simplest methods, the main idea of FGSM is to perturb the data along the sign of the gradient to increase the perturbed linear output, which can be expressed as follows [Goodfellow et al., 2015]:

$$\widetilde{\boldsymbol{x}} = \boldsymbol{x} + \epsilon \, \mathrm{sign}(\nabla_{\boldsymbol{x}} \mathcal{L}(f(\boldsymbol{x}, y))). \tag{5}$$

**PGD**. PGD is an iterative version of FGSM, which offers a more powerful attack [Madry et al., 2018] and is convinced to reasonably approximate the optimal attack. The iteration can be summarized as:

$$\widetilde{\boldsymbol{x}}^k = \Pi_\epsilon \{\boldsymbol{x}^{k-1} + \alpha \, \mathrm{sign}(\nabla_{\boldsymbol{x}} \mathcal{L}(f(\boldsymbol{x}^{k-1}, y)))\}, \tag{6}$$

where $k$ denotes the number of the iteration step and $\alpha$ is the step size of each iteration. $\Pi_\epsilon$ denotes that the data in each iteration should be projected onto the space of the $l_p$ ball around clean data $\boldsymbol{x}$ w.r.t. $\epsilon$. Apart from these popular gradient-based attacks, RFGSM can be viewed as a randomized version of FGSM [Tramèr et al., 2018], and BIM is a kind of iterative attack similar to PGD [Kurakin et al., 2017]. All these methods are exploited to verify the vulnerability of SNN in this work. By applying differentiable approximations in back-propagation, the gradient-based attacks can also threaten SNN. In this work, we execute white-box (WB) and black-box (BB) attacks for scenarios where attackers have knowledge of or no knowledge of the model.

Without specific instructions, we set $\epsilon$ to $8/255$ for all methods for the purpose of testing. For iterative methods like PGD and BIM, the attack step $\alpha = 0.01$, and the step number is 7.

## 2.3 Defense Methods

The earlier research on ANNs has discovered that adversarial attacks can cause the amplification of activation magnitude [Szegedy et al., 2014]. They measured the distance between clean and perturbed activation and suggested an analysis framework based on Lipschitz analysis:

$$\|\boldsymbol{a}^{l+1} - \widetilde{\boldsymbol{a}}^{l+1}\|_2 \leq \mathrm{Lip}_l \|\boldsymbol{a}^l - \widetilde{\boldsymbol{a}}^l\|_2, \tag{7}$$

where $\mathrm{Lip}_l$ is the Lipschitz constant for layer $l$. By penalizing the Lipschitz constant, the distortion of input is stabilized [Cisse et al., 2017]. Using exact estimation of Lipschitz bounds to certify the robustness of ANNs has been popular [Arjovsky et al., 2017, Fazlyab et al., 2019, Weng et al., 2018, Miyato et al., 2018]. Yet there is no work currently working on the Lipschitz bound of SNNs.

Apart from the analysis framework, adversarial training is the most powerful tool for defense, which is defined as a saddle point problem [Madry et al., 2018]:

$$\arg \min_{\boldsymbol{W}} \mathbb{E} \left[ \max_{\boldsymbol{\delta}} \mathcal{L}(f(\boldsymbol{x} + \boldsymbol{\delta}; \boldsymbol{W}), y) \right], \tag{8}$$

where the maximization process can be accomplished by applying different attack methods. By utilizing the adversarial input, the adversarial training learns to classify adversarial examples correctly. All these methods are based on deep ANNs, which are locally differentiable and have well-defined derivatives. In this paper, we aim to construct an analysis framework for non-differentiable SNNs and strengthen the adversarial training.

# 3 Vulnerability of Spiking Neural Networks

## 3.1 Spiking Neural Network under Attack

Although SNNs are more robust than ANNs under some conditions, most SNNs are still vulnerable when effective adversarial attack methods are applied. Previous works have experimentally demonstrated that gradient attack methods such as FGSM can be applied to SNNs [Sharmin et al., 2019, 2020]. However, because of the non-differentiable property of spiking neurons, the gradients obtained by backpropagation are not necessarily accurate, which may cause inefficient attacks. Thus, this defensive nature of SNNs can be thought of as obfuscated gradients [Athalye et al., 2018], which may create a false sense of security about SNNs.

Therefore, in this paper, we first reconsider the attacks for SNNs by utilizing a combination of attack methods and gradient approximations. We follow the Backward Pass Differentiable Approximation (BPDA) technique [Athalye et al., 2018] to overcome the obfuscated gradients. The key idea of the BPDA is to use a differentiable approximation in the backward pass while the forward pass unchanged. Similar differentiable approximation techniques have been used for training SNNs. Based on these, we design, compare, and summarize different differentiable approximations for SNNs.

## 3.2 Differentiable Approximation for Spiking Neural Networks

**Conversion-based Approximation.** The Conversion-based Approximation (CBA) for SNN was first proposed by Sharmin et al. [2019]. Since an SNN can be converted from an ANN [Rueckauer et al., 2017, Han et al., 2020, Deng and Gu, 2021, Ding et al., 2021, Bu et al., 2022a,b], the adversarial examples can be generated from an ANN with shared weights and bias from the source SNN. However, this method approximates the spiking neuron using the ReLU activation function on both forward pass and backward pass, which is contrary to the idea of the BPDA algorithm and is proved to be ineffective [Song et al., 2018].

**Backward Pass Through Time.** The most commonly used differentiable approximation is the Backward Pass Through Time (BPTT) with surrogate gradients [Neftci et al., 2019, Fang et al., 2021b]. In this method, the non-differentiable neuron fire function is replaced by a differentiable function on the backward pass. By combining Eq. 1–Eq. 3, the backward pass can be described as:

$$\frac{\partial L}{\partial \boldsymbol{s}^l(t)} = \frac{\partial L}{\partial \boldsymbol{s}^{l+1}(t)} \frac{\partial \boldsymbol{s}^{l+1}(t)}{\partial \boldsymbol{m}^{l+1}(t^-)} \frac{\partial \boldsymbol{m}^{l+1}(t^-)}{\partial \boldsymbol{s}^l(t)} \tag{9}$$

$$+ \frac{\partial L}{\partial \boldsymbol{s}^l(t+1)} \frac{\partial \boldsymbol{s}^l(t+1)}{\partial \boldsymbol{m}^l(t+1^-)} \frac{\partial \boldsymbol{m}^l(t+1^-)}{\partial \boldsymbol{m}^l(t)} \frac{\partial \boldsymbol{m}^l(t)}{\partial \boldsymbol{s}^l(t)}. \tag{10}$$

Table 1: Performance comparison between differentiable approximations (CBA/BPTR/BPTT).

| Architecture | Dataset | CLEAN | FGSM | RFGSM | PGD |
|---|---|---|---|---|---|
| VGG-11 | CIFAR-10 | 93.06 | 54.34/10.59/12.78 | 72.12/27.28/22.09 | 37.30/00.10/00.04 |
| WRN-16 | CIFAR-10 | 94.38 | 63.32/16.46/14.13 | 77.42/15.46/10.17 | 59.09/00.00/00.00 |
| VGG-11 | CIFAR-100 | 73.33 | 41.80/06.10/05.30 | 57.73/11.95/08.60 | 38.65/00.18/00.02 |
| WRN-16 | CIFAR-100 | 75.33 | 37.68/07.94/07.62 | 54.16/05.99/04.80 | 43.87/00.05/00.00 |

The final result $\frac{\partial L}{\partial \boldsymbol{s}^0(t)}$ can be calculated by recursively calculating this equation across both layers and time-steps. And since we use the constant input coding for SNNs, the gradient of the image is exactly $\frac{\partial L}{\partial \boldsymbol{s}^0(t)}$. The non-differentiable part $\frac{\partial \boldsymbol{s}^{l+1}(t)}{\partial \boldsymbol{m}^{l+1}(t^-)}$ is replaced by a surrogate gradient function to get a smooth backward pass. The backward pass through time technique uses a similar gradient approximation as the Spatio-Temporal-Backward-Propagation training algorithm [Wu et al., 2018b]. Since the gradient generated from this method is useful for training, it is very likely to generate effective gradients.

**Backward Pass Through Rate.** Another differentiable approximation is the Backward Pass Through Rate (BPTR). In this method, the backward pass takes the derivative directly from the average firing rate of the spiking neurons between layers. We consider neurons at each time-step equivalent, and the derivative is determined by the average firing rate.

$$\forall t \in \{1, 2, \cdots, T\}, \quad \frac{\partial L}{\partial \boldsymbol{s}^l(t)} = \frac{\partial L}{\partial \boldsymbol{s}^{l+1}(t)} \frac{\partial \frac{1}{T} \sum_{i=0}^{T} \boldsymbol{s}^{l+1}(i)}{\partial \frac{1}{T} \sum_{i=0}^{T} \boldsymbol{s}^l(i)}. \tag{11}$$

Since the relationship of the firing rates in adjacent layers for non-leaky IF neuron is nearly linear [Sengupta et al., 2019], the gradients of $\partial \frac{1}{T} \sum_{i=0}^{T} \boldsymbol{s}^{l+1}(i) / \partial \frac{1}{T} \sum_{i=0}^{T} \boldsymbol{s}^l(i)$ can be approximated using the straight-through estimator [Bengio et al., 2013]. Similar to Lee et al. [2020], here we use the constant $\frac{1}{T}$ to approximate the value of $\partial \frac{1}{T} \sum_{i=0}^{T} \boldsymbol{s}^{l+1}(i) / \partial \frac{1}{T} \sum_{i=0}^{T} \boldsymbol{s}^l(i)$ at Eq. 11. The backward pass of the complete neuronal dynamic is approximated by one single function, and the gradient will not accumulate through time-steps. As the forward pass still follows the rule of the spiking neurons, the obtained gradients will be more accurate than the conversion-based attack.

### 3.3 Effective Attack with Backward Pass through Time and Rate

To compare the effectiveness of the above three differentiable approximation techniques in constructing gradient-based attacks for SNNs, we applied combinations of three gradient-based methods (FGSM, RFGSM, PGD) and three differentiable approximation techniques (CBA, BPTR, BPTT) to two baseline models. The two models, VGG-11 and WideResNet-16, are trained on the CIFAR dataset using BPTT with no additional defenses. The number of time-steps is set to $T = 8$.

As shown in Tab. 1, the CBA is the most ineffective differentiable approximation technique to construct attacks for SNNs. The performance of all models under attack using CBA is significantly higher than that of models under attack using BPTR and BPTT. As we have discussed in Sec. 3.2 when using CBA, the changes in both forward pass and backward pass cause inaccuracy in the generated gradient. Both the BPTT and BPTR methods, combined with different attack methods, can significantly reduce the performance of the given model. Among all single-step attack combinations, FGSM(BPTR), RFGSM(BPTT), FGSM(BPTT), and RFGSM(BPTT) can generate efficient and effective adversarial examples for all the testing models. Among all multi-step attacks, the PGD(BPTT) always beats other attack combinations. These results suggest different properties between BPTT and BPTR. The BPTT approximation contains extra temporal information that is useful when using multi-step attacks, while the BPTR approximation saves more computation resources when back-propagating through firing rates and can get comparable results on single-step attacks.

# 4 Methods: Perturbation Analysis and Regularized Adversarial Training

The previous section identifies threats from effective differentiable approximations combined with gradient-based attacks. In this section, we will give a theoretical perturbation analysis under SNN attacks and propose a regularized adversarial training scheme.

## 4.1 Perturbation Analysis for Spike Representation

Current perturbation analysis of ANN considers the distance of continuous activation $\|\boldsymbol{a}^l - \widetilde{\boldsymbol{a}}^l\|$ and the rectified linear outputs $\|f(\boldsymbol{a}^l) - f(\widetilde{\boldsymbol{a}}^l)\|$. Kundu et al. [2021b] used this rate-based distance to bridge the robustness of ANN and SNN. Spike trains in SNN not only contain rate information but also have a temporal structure. To evaluate the distance in the spike train space, various kernel methods are proposed for neuronal identification and encoding [Weng et al., 2018]. Inspired by these works, we propose to model the distortion of the spike response using the spike train distance, which may bridge the robustness of SNN to the discovery of neuroscience and is also sensitive to the change of both firing rate and temporal information. Denote the output spike train of the $l^{th}$ layer as $\boldsymbol{S}^l = \{\boldsymbol{s}^l(t) | t = 1, 2, \cdots, T\} \in \chi^{T \times N_l}$ ($\chi \in \{0, 1\}$), where $T$ is the number of time-steps and $N_l$ is the number of neurons in layer $l$. Then the perturbation distance can be formulated as ($p \geq 1$):

$$D_p(\boldsymbol{S}^l, \widetilde{\boldsymbol{S}}^l) = \|\boldsymbol{S}^l - \widetilde{\boldsymbol{S}}^l\|_{p,p} = \left( \sum_{t=1}^{T} \|\boldsymbol{s}^l(t) - \widetilde{\boldsymbol{s}}^l(t)\|_p^p \right)^{1/p}, \tag{12}$$

where $\| \cdot \|_{p,p}$ denotes the $l_p$ entry-wise matrix norm and $\| \cdot \|_p$ denotes the $l_p$ vector norm. $\widetilde{\boldsymbol{S}}^l$ is the perturbed version of $\boldsymbol{S}^l$. Since the $p$ power of spike activation values (0 and 1) are themselves, without loss of generality we set $p = 2$. In Theorem 1, we obtain a Lipschitz constant for the spike train distance.

**Theorem 1.** *Given an L-layered SNN intended to inference $T$ time-steps with $\theta$ as threshold, suppose that there are $N_l$ neurons in layer $l$ for $l = 1, 2, \cdots, L$. $\boldsymbol{W}^l \in \mathbb{R}^{N_l \times N_{l-1}}$. For layer $l$, it satisfies:*

$$D_2(\boldsymbol{S}^l, \widetilde{\boldsymbol{S}}^l)^2 \leq \frac{1}{\theta^2} \Lambda^{l2} D_2(\boldsymbol{S}^{l-1}, \widetilde{\boldsymbol{S}}^{l-1})^2 + \Gamma^l, \tag{13}$$

*where $\Lambda^l$ is a Lipschitz constant and $\Gamma^l$ is a constant for layer $l$, which can be expressed as:*

$$\Lambda^l = \sup_{\boldsymbol{s} \neq 0, \boldsymbol{s} \in \psi^{N_{l-1}}} \frac{\|\boldsymbol{W}^l \boldsymbol{s}\|_2}{\|\boldsymbol{s}\|_2}, \tag{14}$$

$$\Gamma^l = \frac{N_l T(T+1)}{\lambda} \left[ \frac{\gamma^l}{\theta} + \left( \frac{\gamma^l}{\theta} \right)^2 \right], \tag{15}$$

*where $\gamma^l = \sup_{\boldsymbol{s} \neq 0, \boldsymbol{s} \in \chi^{N_{l-1}}} \|\boldsymbol{W}^l \boldsymbol{s}\|_\infty + \sup_{\boldsymbol{s} \neq 0, \boldsymbol{s} \in \chi^{N_{l-1}}} \| -\boldsymbol{W}^l \boldsymbol{s}\|_\infty. \chi = \{0, 1\}, \psi = \{-1, 0, 1\}.$*

In Eq. 14, $\Lambda^l$ is referred to as the **spiking Lipschitz constant**. Since the inter-layer spike signals are not purely linear and the effect of the spiking generation mechanism cannot be neglected, the inequality relationship expressed by Eq. 13 is not exactly the same as the definition of the classical Lipschitz constant [O'Searcoid, 2006]. Nonetheless, this does not affect our understanding of the amplification effect of spike distance. On the right side of Eq. 13, there is an additional constant $\Gamma^l$ related to the weight. The appearance of $\Gamma^l$ is associated with the spike generation and the bounding of the current. The determination of $\Gamma^l$ adopts a very loose constraint (see our proof in the Appendix), and its effect on the spike distance is additive, so we will still focus on how to constrain the spiking Lipschitz constant while not magnifying the additive item excessively.

## 4.2 Constraining Spiking Lipschitz Constant

Here we explain why SNNs are thought to be more robust than ANNs and how to constrain the spiking Lipschitz constant in Eq. 13. Szegedy et al. [2014] has concluded that the upper bound of the Lipschitz constant for ReLU-based layers is the largest singular value of the weight matrix, i.e., $\sup_{\boldsymbol{x} \neq 0} \frac{\|\boldsymbol{W}^l \boldsymbol{x}\|_2}{\|\boldsymbol{x}\|_2}$ for layer $l$. The difference between the classical Lipschitz constant and the

spiking Lipschitz constant lies in the vector domain. The classical Lipschitz constant is the attained supremum for any vector $x \neq 0$. In the case of the SNN layer, its Lipschitz constant is constrained in the space $\{x \neq 0, x \in \psi^{N_{l-1}}, \psi = \{-1, 0, 1\}\}$, a subspace of $x \neq 0$. Inspired by this, we can derive an inequality between the classical Lipschitz constant and the spiking Lipschitz constant, which is presented in Proposition 1 (The detailed proof is in the Appendix).

**Proposition 1.** *Given a weight matrix with real values $W^l \in \mathbb{R}^{N_l \times N_{l-1}}$. $\psi = \{-1, 0, 1\}$. It satisfies:*

$$\Lambda^l = \sup_{s \neq 0, s \in \psi^{N_{l-1}}} \frac{\|W^l s\|_2}{\|s\|_2} \leq \|W^l\|_2 = \sigma^{\max}(W^l), \tag{16}$$

*where $\|W^l\|_2$ is the induced $l_2$ matrix norm, and $\sigma^{\max}(W^l)$ is the largest singular value of $W^l$.*

According to Eq. 16, the spiking Lipschitz constant $\Lambda^l$ is less than $\sigma^{\max}(W^l)$. The primary goal is to constrain the training through a weight regularizer. However, $\Lambda^l$ is hard to estimate, whereas there are numerous ways to modulate the spectral norm $\sigma^{\max}(W^l)$ [Miyato et al., 2018, Cisse et al., 2017, Yoshida and Miyato, 2017]. Thus, we choose to adopt spectral norm regularization into our proposed training scheme. The target of regularization is to control the spectral norm to approach 1:

$$\text{Goal: } \Lambda^l \to 1 \quad \text{Implementation: } \sigma^{\max}(W^l) \to 1. \tag{17}$$

Such that according to Eq. 16, we can also meet the criterion that $\Lambda^l$ is less than 1 for each layer $l$. We note that the constraint to $\|W^l\|_2$ can also contribute to limiting $\gamma^l$, which is presented in Proposition 2 (The detailed proof is in the Appendix).

**Proposition 2.** *Given a weight matrix with real values $W^l \in \mathbb{R}^{N_l \times N_{l-1}}$. $\chi = \{0, 1\}$. It satisfies:*

$$\gamma^l = \sup_{s \neq 0, s \in \chi^{N_{l-1}}} \|W^l s\|_\infty + \sup_{s \neq 0, s \in \chi^{N_{l-1}}} \| - W^l s\|_\infty \leq 2\|W^l\|_\infty \leq 2\sqrt{N_{l-1}}\|W^l\|_2, \tag{18}$$

*where $\|W^l\|_p$ is the induced $l_p$ matrix norm.*

In Proposition 2, the relationship between $l_\infty$ norm and $l_2$ norm is based on the norm inequality theory. The item in $\gamma^l$ defined in a constrained space, on the other hand, is strictly less than the $l_\infty$ norm of $W^l$. Thus, we can conclude $\gamma^l$ has an upper bound determined by scaled $\sigma^{\max}(W^l)$. Controlling $\sigma^{\max}(W^l)$ is sufficient to limit $\Gamma^l$ as $\Gamma^l$ monotonously increases with the increase of $\gamma^l$.

### 4.3 Regularized Adversarial Training (RAT)

**Application of regularization (REG)**. Once the target of regularization is set to control the matrix norm of weights, the next question is how to find a good regularizer to assist training. By setting the threshold of spiking neurons to 1, the amplification of spike distance is mainly related to $\Gamma$. Given that SNN frequently suffers from gradient problems during training [Wu et al., 2018b], we eventually settle on an orthogonal regularization method that can help prevent the network from vanishing gradients [Lin et al., 2021]. Singular values are naturally equal to 1 when the weight matrix row is orthogonal. Therefore, we propose to project the updating weights to the target of the orthogonal matrix at a rate defined by $\beta$ [Cisse et al., 2017]:

$$\forall l = 1, 2, \cdots, L, \quad W^l \leftarrow \Pi_{W^T W = I}\left(W^l\right) \tag{19}$$

$$\Pi_{W^T W = I}(W) = W - \beta\left(WW^T W - W\right) \tag{20}$$

For convolutional layers that have weight matrices of 4 dimensions $W \in \mathbb{R}^{C_{out} \times C_{in} \times k \times k}$, these matrices should be first reorganized into 2-dimensional matrices in $\mathbb{R}^{C_{out} \times (C_{in} \times k \times k)}$ to accomplish the update, where $k$ is the kernel size and $C_{in}, C_{out}$ are numbers of input and output channels.

**Generalizing strong SNN adversarial examples (MIX)**. As Sec. 3 suggested, SNNs are also vulnerable under different approximations of the backward pass. To equip the trained SNN with even better invulnerability, we argument the training dataset with stronger adversarial perturbations based on our observations. The analyses and results in Sec. 3 imply that the single-step FGSM and RFGSM can provide us with adequate adversaries at a relatively low computational cost. It is worth noting that RFGSM is recognized as an effective substitution for PGD in adversarial learning [Wong et al.,

2019]. The BPTT and BPTR approximations further allow the dataset to extend to its $l_\infty$ balls with a strong adversary compared with the CBA. Hence, our model is trained with adversarial examples randomly sampled from {FGSM(BPTT), RFGSM(BPTT), FGSM(BPTR), RFGSM(BPTR)}. Each mini-batch of images are perturbed by a single choice from the four methods with equal probability before being fed into the network. The network learns to generalize in the mixture of heterogeneous neighbourhoods defined by the four adversaries. For better network training performance, we choose the mainstream method where the network is updated by the gradients produced from BPTT with surrogate functions [Neftci et al., 2019, Zheng et al., 2021, Fang et al., 2021a].

## 5 Experiments

### 5.1 Experimental Setup

We validate our proposed robust SNN training scheme on the image classification tasks, where the CIFAR-10 and CIFAR-100 datasets are used. We train the SNN version of VGG-11 and WideResNet-16 (with widening factor set as 4) for both datasets with $T = 8$. In addition, we set $\beta = 0.001$ and $0.004$ for VGG-11 and WideResNet-16, respectively. The perturbation boundary $\epsilon$ is set to 2/255 when training models. Detailed implementation is referred to in Appendix.

The images are directly fed into SNNs. We include several gradient-based attack methods to evaluate the adversarial performance thoroughly: FGSM [Goodfellow et al., 2015], RFGSM [Tramèr et al., 2018], PGD [Madry et al., 2018], BIM [Kurakin et al., 2017]. Gaussian noise (GN) is also used to test the performance against random perturbation with the same $\epsilon$.

### 5.2 Results

**Performance under attacks**. Tab. 2 reports the performance of our proposed RAT scheme. All gradient attacks are combined with powerful SNN attacks (BPTT, BPTR). The classification accuracy in brackets is the accuracy without the proposed training scheme. Black-box attacks are marked with "*" in the table. It is observed that for all the attack methods, our RAT can improve the model robustness, which is reflected in the improvement of accuracy. Compared with VGGs, Vanilla WideResNets are more vulnerable to RFGSM attacks. However, armed with RAT, it is even more robust than VGGs. Black-box attacks are almost ineffective against RAT-trained models. For stronger white-box iterative attacks, our RAT improves the robustness from being almost completely misclassified. For example, the VGG-11 model increases its accuracy by 34.76% under the PGD(BPTR) attack after being trained with RAT on the CIFAR-10 dataset.

**Performance with larger $\epsilon$.** We plot the accuracy of the white-box and black-box scenarios under the PGD(BPTT) attack in Fig. 1. For white-box attacks, both models converge to almost zero after $\epsilon = 4/255$, while RAT-trained models drop their accuracy with increasing $\epsilon$. Black-box attacks are weaker than white-box ones. From Fig. 1(b) and (d), the accuracy of RAT-trained models decreases comparably slowly to that of vanilla models.

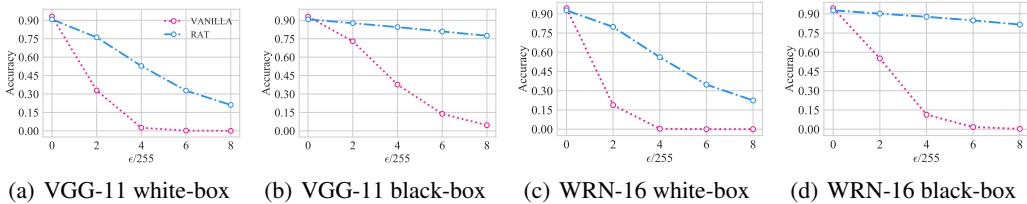

| (a) VGG-11 white-box | (b) VGG-11 black-box | (c) WRN-16 white-box | (d) WRN-16 black-box |

Figure 1: Performance under PGD(BPTT) attacks.

### 5.3 Ablation Studies

The proposed RAT scheme in Sec. 4.3 is composed of a regularizer to control the spiking Lipschitz constant and mixed adversarial neighbourhoods for adversarial training. We conduct ablation studies based on VGG-11 trained with the CIFAR-10 dataset. The results are shown in Tab. 3. The attack

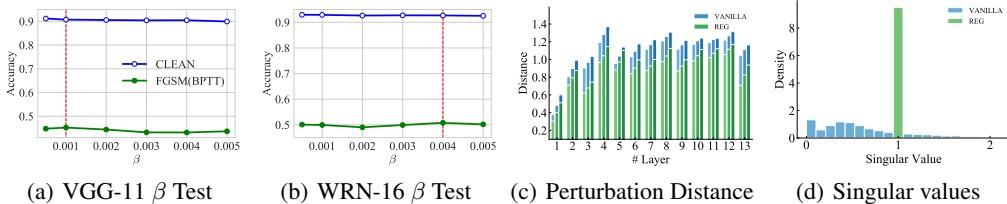

|  | (a) VGG-11 $\beta$ Test | (b) WRN-16 $\beta$ Test | (c) Perturbation Distance | (d) Singular values |

Figure 2: Effects of Regularized Adversarial Learning.

methods are all BPTT attacks. When trained without RAT, the performances under single-step FGSM and RFGSM are lower than 25%, and the accuracies of PGD and BIM are nearly zero. With only the application of the regularization, the performance of FGSM and RFGSM increase to 26.6% and 40.9%. The single deployment of mixed adversarial training significantly improves the robustness as the combination of heterogeneous neighbourhoods enhances data generalization. When both the regularizer and the adversarial training are used, the model gets the best robustness. These results indicate that each component in RAT alone can improve the model's adversarial performance, and the robustness of the model is greatly improved after they are combined.

### 5.4 Effects of Regularized Adversarial Learning

We test the sensitivity of the orthogonal projection rate $\beta$ on the CIFAR-10 dataset with VGG-11 and WideResNet-16 structures. As shown in Fig. 2(a) and (b), $\beta$ has little effect on clean accuracy. For the two architectures, 0.001 and 0.004 give the best accuracy under the FGSM(BPTT) attack, respectively. To validate the effect of the regularization, the perturbation distance of each neuron layer in WideResNet-16 is visualized. In Fig. 2(c), the colors range from light to dark denote $\epsilon = 2, 4, 8$. It can be seen that the distances of the regularized model are consistently less than those of the vanilla model, which indicates that regularization inhibits the amplification of the spike distance. Fig. 2(d) presents the distribution of singular values of the WideResNet-16 model. The regularization makes singular values of weights to be aggregated at 1 as expected.

Table 2: Performance of the proposed RAT under different attacks.

| Attack | VGG/CIFAR10 | WRN/CIFAR10 | VGG/CIFAR100 | WRN/CIFAR100 |
|---|---|---|---|---|
| Clean | 90.74(93.06) | 92.69(94.38) | 70.89(73.33) | 69.32(75.33) |
| GN | 90.31(90.77) | **91.35**(83.08) | 66.01(66.77) | **66.28**(49.86) |
| FGSM(BPTR) | **51.77**(10.59) | **55.70**(16.46) | **28.24**(6.10) | **31.75**(07.94) |
| FGSM(BPTT) | **45.23**(12.78) | **50.78**(14.13) | **25.86**(5.30) | **28.08**(07.62) |
| RFGSM(BPTR) | **70.69**(27.28) | **73.32**(15.46) | **43.50**(11.95) | **45.68**(05.99) |
| RFGSM(BPTT) | **64.61**(22.09) | **69.30**(10.17) | **38.72**(8.60) | **40.51**(04.80) |
| PGD(BPTR) | **34.86**(00.10) | **36.36**(00.00) | **18.15**(0.18) | **19.32**(00.05) |
| PGD(BPTT) | **21.16**(00.04) | **22.71**(00.00) | **10.38**(00.02) | **11.31**(00.00) |
| BIM(BPTR) | **33.29**(00.08) | **34.92**(00.00) | **17.81**(00.17) | **18.89**(00.05) |
| BIM(BPTT) | **18.64**(00.03) | **19.58**(00.00) | **09.50**(00.03) | **10.25**(00.00) |
| FGSM(BPTR)* | **75.71**(32.11) | **77.66**(25.25) | **48.54**(18.31) | **50.11**(13.95) |
| FGSM(BPTT)* | **73.76**(28.51) | **76.96**(24.49) | **47.25**(16.81) | **50.39**(13.50) |
| RFGSM(BPTR)* | **83.79**(56.82) | **86.45**(31.71) | **59.67**(32.66) | **60.73**(17.80) |
| RFGSM(BPTT)* | **83.35**(49.91) | **85.90**(26.68) | **58.71**(30.06) | **60.06**(15.71) |
| PGD(BPTR)* | **77.15**(12.02) | **82.81**(00.94) | **55.44**(10.54) | **59.23**(03.31) |
| PGD(BPTT)* | **77.43**(04.61) | **82.20**(00.31) | **54.11**(07.04) | **58.28**(02.53) |
| BIM(BPTR)* | **76.16**(10.57) | **81.16**(00.88) | **54.10**(09.53) | **56.52**(03.53) |
| BIM(BPTT)* | **75.89**(04.00) | **79.93**(00.29) | **52.60**(06.42) | **55.86**(02.49) |

Table 3: Ablation study with VGG-11 on CIFAR-10.

| MIX | REG | Clean | FGSM | RFGSM | PGD | BIM |
|-----|-----|-------|------|-------|-----|-----|
| × | L2 | 93.06 | 12.78 | 22.09 | 0.04 | 0.03 |
| × | ✓ | 91.09 | 26.60 | 40.89 | 0.85 | 0.55 |
| ✓ | L2 | 92.04 | 37.49 | 60.11 | 16.63 | 15.28 |
| ✓ | ✓ | 90.74 | 45.23 | 64.61 | 21.16 | 18.64 |

Table 4: Comapare with state-of-the-art work on adversarial robustness of SNN.

| BPTT Attack | FGSM | PGD | Clean |
|-------------|------|-----|-------|
| Sharmin et al. [2020] | 15.50 | 6.30 | 64.40 |
| Kundu et al. [2021b] | 22.00 | 7.50 | 65.10 |
| Vanilla | 5.30 | 0.02 | 73.33 |
| **Our work** | **25.86** | **10.38** | 70.89 |

## 5.5 Comparison with State-of-the-art Work on Adversarial Robustness of SNN

We compare our methods with the state-of-the-art models and report the results in Tab. 4. The evaluation is based on the VGG-11 experiments on the CIFAR-100 dataset. The noise budget has been fixed to $\epsilon = 8/255$ for FGSM and $\alpha = 0.01, step = 7$ for PGD. The attack is based on the surrogate gradient produced by BPTT. The performance of accuracy attacked by FGSM is 25.86% for our work, higher than that proposed by Sharmin et al. [2020] (15.5%) and Kundu et al. [2021b] (22.0%). Apart from that, our clean accuracy (70.89%) is higher than that proposed by Sharmin et al. [2020] (64.4%) and Kundu et al. [2021b] (65.1%). This implies that our proposed methods can bring better generalization compared to the SOTA robust models.

It is worth noting that although our training algorithm improves the robustness of SNNs, it comes at extra costs. The cost is mainly reflected in the training time. First, the regularization of the weights is computed every update. Solutions to reduce the time consumption of regularization include sampling fewer weights for regularizing, or reducing the number of regularization updates.

Besides, the generation of adversarial noise, which is also included in Kundu et al. [2021b], costs some time. Adversarial learning is a common scheme to improve robustness, and generating adversarial examples using only BPTT differentiable approximation in SNN is a time-consuming operation. Our algorithm mitigates the increase in training time by mixing in a faster yet efficient BPTR approximation. To verify this, we evaluate the computational time of adversarial testing, detailed settings are referred to in Appendix. The results show that: BPTR is almost as efficient as CBA, and BPTT costs nearly $3\times$ of what CBA and BPTR take to complete testing.

## 6 Conclusions and Discussions

In this work, we are the first to give a theoretical Lipschitz analysis of perturbation for hardware-friendly SNNs. SNNs are more vulnerable due to their diverse and feasible gradient methods. Therefore, we propose a specialized regularization adversarial training scheme for SNNs. Our experiments demonstrate that models trained on this scheme can obtain much robustness, especially in black-box attacks. We believe this work will pave the way for SNNs in energy-efficient as well as safety-critical applications. Besides, recent works have shown that SNN can achieve good results without BN [Kundu et al., 2021a]. Note that BN are included in our model, which may be harmful to the robustness [Wang et al., 2022]. Thus, valuable future research directions will include how to train robust SNNs while getting rid of the adverse effects of BN.

## 7 Acknowledgements

We thank Yujia Liu for valuable discussions. This work was supported by the National Natural Science Foundation of China Grants 62176003 and 62088102.

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
