# OpenReview forum: "SNN-RAT: Robustness-enhanced Spiking Neural Network through Regularized Adversarial Training"
_NeurIPS.cc/2022/Conference — NeurIPS 2022 Accept_

### Official Review · Reviewer_PjMN · 2022-06-30

**Rating:** 6
**Confidence:** 4
**Soundness:** 3 good
**Presentation:** 3 good
**Contribution:** 3 good

**Summary:**

This paper provides a theoretical analysis of the SNN robustness against adversarial perturbation. In this regard, a regularized adversarial training method for SNN has been proposed. The regularization is based on the spiking Lipschitz constant. The results conducted on various benchmarks demonstrate that the proposed training scheme achieves better robustness compared to the vanilla adversarial training.

**Questions:**

1.	In Section 1: “it is necessary to improve the adversariality of SNNs”. Do you mean to improve the adversarial robustness of SNNs?
2.	In Section 1: “The update amount of weights is the key to constructing adversarial attacks.” Please rewrite this sentence. The adversarial attacks should update the input intensities, rather than the weights.
3.	What is the noise budget used in the attacks for the results reported in Table 1?
4.	The proposed regularization method described in Section 4 is strongly based on previous works. It is recommended to explain more clearly the novel features.
5.	In Section 5, the results of the proposed method have been compared only to the vanilla adversarial training. If possible, the comparison with other methods among the related works should be included.
6.	The experiments are conducted only on static data, while SNNs are commonly used also on event-based data. Therefore, it is recommended to extend the experiment set including results on event-based datasets.

**Limitations:**

The limitations and societal impact have been discussed by the authors in the supplementary material.

**Strengths And Weaknesses:**

Strengths:
1.	The proposed idea is original and relevant to the NeurIPS community.
2.	The results show an advancement in the state-of-the-art.

Weaknesses:
1.	Some key concepts need better clarification. See the specific questions below.
2.	The experiments should be extended using event-based datasets.

---

> ### Author Response · Authors · 2022-08-02
> **Response to Reviewer PjMN**
>
> We appreciate the reviewer for the advice. We are grateful that you find our paper original and exhibit advanced performance. We would like to address your concerns and answer your questions here.
>
> ## 1. In Section 1: “it is necessary to improve the adversariality of SNNs”. Do you mean to improve the adversarial robustness of SNNs?
>
> Yes, we mean to improve the adversarial robustness of SNNs. As the expression may cause confusion to readers, we have rewritten it in the revised paper.
>
> ## 2. In Section 1: “The update amount of weights is the key to constructing adversarial attacks.” Please rewrite this sentence. The adversarial attacks should update the input intensities, rather than the weights.
>
> Thanks for pointing it out. We have rewritten the sentence to *''the key to constructing SNN gradient-based attacks is back-propagation, which is the same as that of ANNs''* to avoid confusion.
>
> ## 3. What is the noise budget used in the attacks for the results reported in Table 1?
>
> The noise budget is 8/255 for FGSM and (alpha=0.01, step=7) for PGD. To clearly clarify the noise budget, we have moved the declaration to Section 2.2, which introduces gradient-based methods:
> *''Without specific instructions, we set $\epsilon$ to $8/255$ for all methods for the purpose of testing. For iterative methods like PGD and BIM, the attack step $\alpha=0.01$, and the step number is 7. ''*
>
> ## 4. The proposed regularization method described in Section 4 is strongly based on previous works. It is recommended to explain more clearly the novel features.
>
> You have raised an important concern. We would like to clarify that our novelty on Lipschitz is mainly reflected in the theoretical derivation and implication. The calculation scheme of Lipschitz constant of ANNs cannot be directly applied to SNNs with discrete activation and time-series processing capability, which forms difficulty in determining Lipschitz constant of SNNs. As described in Section 4, we theoretically give a bound on the Lipschitz constants for SNNs using a spike distance. The biggest differences between previous ANN work and our work lie in two points.
> 1. The bound space for computing matrix norm is very different from that of ANN, which is supported by our Theorem 1;
> 2. The spiking Lipschitz has an upper bound over traditional Lipschitz, which is supported by our Proposition 1. Based on these results, we adopt orthogonal regularization to control the spiking Lipschitz.
>
> We believe our theoretical work on spiking Lipschitz will raise the focus of the community on the theoretical bounds of SNN robustness.
>
> ## 5. In Section 5, the results of the proposed method have been compared only to the vanilla adversarial training. If possible, the comparison with other methods among the related works should be included.
>
> Thanks for pointing it out. Please refer to the Section of To All Reviewers. We have included the comparison with SOTA models in the revised paper (please refer to the appendix).
>
> ## 6. The experiments are conducted only on static data, while SNNs are commonly used also on event-based data. Therefore, it is recommended to extend the experiment set including results on event-based datasets.
>
> You have raised an interesting concern. SNN is indeed known to be commonly used in event-based data. We would like to point out that event data consists of discrete spikes while the adversarial methods (FGSM, BIM, etc.) produce floating-point attack. How to perform attack with different noise budgets on event-based data has not yet been thoroughly discussed. Hence, this paper focuses more on the floating-point data type static images with mature attack methods. We are glad to investigate the robustness of event-based data further in future work.

---

> ### Author Response · Authors · 2022-08-07
> **Thanks for the precious time and we hope to see if there is any further concern**
>
> Dear Reviewer PjMN,
>
> Thank you for the detailed feedbacks and constructive suggestions. As the discussion period will end soon, we would like to kindly ask if our previous response clarifies your concerns and if there are any further questions that we could answer to facilitate the review process. Thanks a lot for your time!

---

> ### Author Response · Authors · 2022-08-08
> **Thank you for the time and we hope that our response help for your assessment of our work**
>
> Dear Reviewer PjMN,
>
> We notice that your concerns lie in five parts in the previous review.
>
> 1) The first concern is about the ambiguous expressions, for which we have altered the expression.
> 2) The second concern is the absence of experimental parameters, for which we provide the absent noise budget parameters.
> 3) The third concern is about the novel features of Lipschitz, for which we explain that the novel features lie in the theoretical derivations for spiking Lipschitz analysis. We would like to note that the calculation scheme of Lipschitz constant of ANNs cannot be directly applied to SNNs with discrete activation and time-series processing capability, which forms a difficulty in determining Lipschitz constant of SNNs.
> 4) The fourth concern is the comparison with the SOTA works, for which we provide a comparison to the SOTA related works of SNN robustness (see the section of **TO All Reviewers**).
> 5) The fifth concern is about the experiment on event-based datasets, yet how to perform attacks on event data has not been thoroughly discussed, so we would like to extend the experiments in our future work.
>
> We sincerely hope that our feedback could settle and answer your concerns. Also, if you have any further questions or comments, please let us know, and we are glad to give further responses.

---

> > ### Comment · Reviewer_PjMN · 2022-08-09
> > **Response to Authors' rebuttal**
> >
> > Dear Authors,
> >
> > Thanks for answering the reviewers' comments in a clear and comprehensive way.
> >
> > I don't have further questions.

---

### Official Review · Reviewer_BBvf · 2022-07-11

**Rating:** 5
**Confidence:** 3
**Soundness:** 3 good
**Presentation:** 3 good
**Contribution:** 3 good

**Summary:**

This paper proposes Regularized Adversarial Training (RAT), an adversarial training framework to improve the robustness of the Spike Neural Network (SNN). This paper adopts three different gradient approximations (i.e., CBA, BPTR, BPTT) to mitigate the non-differentiable of SNN. At first, to achieve stronger adversaries, this paper augments FGSM, RFGSM, and PGD with the above approximations. Then this paper proposes spiking the Lipschitz constant, a variant of normal Lipschitz constraint that regularizes the model to be Lipschitz smooth. The proposed RAT is evaluated over several popular architectures.

**Questions:**

1. Some key references are missing in this paper, including the robustness of SNN and the Lipschitz constraint in ANN. The authors should make a comprehensive summary of these fields and highlight their contributions.
2. The authors should provide comparisons with other state-of-the-art defense methods on SNN and also the baseline of this work.

**Limitations:**

The authors have addressed the limitations and potential negative societal impact of this work.

**Strengths And Weaknesses:**

**Strengths:**

1. This paper is well-written and easy to follow.
2. The proposed SAT seems very effective over several strong adversaries.

**Weaknesses:**

1. This paper aims to improve the robustness of SNN but didn’t summarize previous efforts on this topic. Besides, this paper didn’t provide comparisons with other related state-of-the-art algorithms, which makes it hard to recognize the contribution and value to this field.
2. Training network with Lipschitz constraints have been welly exploited in ANN. Some key references are missing in this paper. The authors should illustrate the differences between their work and those Lipschitz constraints proposed on ANN.
3. The setting of the ablation study is unclear. In Table 3, the definitions of MIX and REG are not provided. Besides, what is the baseline of this algorithm? Since this work is named Regularized Adversarial Training, I expect the baseline would be normal adversarial training without Lipschitz constraint. However, I couldn’t find it.

---

> ### Author Response · Authors · 2022-08-02
> **Response to Reviewer BBvf**
>
> Thank you for your detailed and insightful comments. We are delighted that you find our paper well-written and seem very effective. We would like to address your concerns and answer your questions here.
>
> ## 1. This paper aims to improve the robustness of SNN but didn’t summarize previous efforts on this topic. Besides, this paper didn’t provide comparisons with other related state-of-the-art algorithms, which makes it hard to recognize the contribution and value to this field.
>
> Thanks for your comments. We would like to clarify that SNN adversarial robustness is a very new research field, and there are still few works focusing on improving the adversarial robustness of SNNs. We have rewritten Section 2.1 (related work on SNN robustness) by adding the literature on SNN robustness therein. SNNs are robust due to input coding, spike communication, potential decay, etc.
>
> Besides, we have added comparisons with these related SOTA algorithms as you suggested. Please refer to the section of **To All Reviewers**. We hope this will increase your recognition of our work.
>
> ## 2. Training network with Lipschitz constraints have been well exploited in ANN. Some key references are missing in this paper. The authors should illustrate the differences between their work and those Lipschitz constraints proposed on ANN.
>
> We agree that training networks with Lipschitz constraints have been well exploited in ANN, and the robustness of ANN can be improved when combined with Lipschitz regularization. Thanks for your suggestion，we have added more related work on Lipschitz constraints in Section 2.3.
>
> We would like to note that the calculation scheme of Lipschitz constant of ANNs cannot be directly applied to SNNs with discrete activation and time-series processing capability, which forms a difficulty in determining Lipschitz constant of SNNs. As described in Section 4, we theoretically give a bound on the Lipschitz constants for SNNs using a spike distance. The biggest differences between previous ANN work and our work lie in two points. 1. The bound space for computing matrix norm differs significantly from that of ANN, which is supported by our Theorem 1; 2. The spiking Lipschitz has an upper bound over traditional Lipschitz, which is supported by our Proposition 1. Based on these results, we adopt orthogonal regularization to control the spiking Lipschitz.
>
> Thus, our main contributions are in the following:
>
> (1) We design and summarize three different gradient approximations (i.e., CBA, BPTR, BPTT) to attack the non-differentiable SNN.
>
> (2) We theoretically give a bound on the Lipschitz constants for SNNs using a spike distance. Our theoretical results show that the spiking Lipschitz differs ANN Lipschitz in the norm space, and it has an upper bound.
>
> (3) Based on our theoretical implication, we propose a regularized adversarial training scheme for SNN, which proves to be effective in our experiments.
>
> We believe our theoretical work on spiking Lipschitz will raise the focus of the community on the theoretical bounds of SNN robustness.
>
> ## 3. The setting of the ablation study is unclear. In Table 3, the definitions of MIX and REG are not provided. Besides, what is the baseline of this algorithm? Since this work is named Regularized Adversarial Training, I expect the baseline would be normal adversarial training without Lipschitz constraint. However, I couldn’t find it.
>
> Thanks for your suggestions. The proposed RAT scheme is composed of ''a regularizer to control the spiking Lipschitz constant'' (abbreviated to REG) and ''mixed adversarial neighbourhoods for adversarial training'' (abbreviated to MIX). So our ablation study is to discuss the effect of the two training components.
>
> We would like to clarify that the result of adversarial training without Lipschitz constraint is provided in Line 3, Table 3 of the manuscript. {AT+no regularization} achieves FGSM-attacked accuracy of (37.49%), which is higher than that of {no AT+regularization} (26.60%), but less than that of {AT+regularization} (45.23%).

---

> > ### Comment · Reviewer_BBvf · 2022-08-09
> > **Thank you for your response**
> >
> > After reading the rebuttal, most concerns are addressed. I will increase my score to borderline accept. The combination of adversarial training with SNN seems promising but the current version lacks theoretical contribution.

---

> ### Author Response · Authors · 2022-08-07
> **Looking forward to feedbacks**
>
> Dear Reviewer BBvf,
>
> Thanks for your thorough initial comments. We really hope to know whether our previous response has addressed your questions and concerns properly. Since it is approaching the end of author-reviewer discussion period, please let us know if you have any further comments, and we are glad to write a follow-up response. Thank you very much!

---

> ### Author Response · Authors · 2022-08-08
> **Thanks for your precious time and we would like to see if there is any further concern and comment**
>
> Dear Reviewer BBvf,
>
> We notice that in your initial review, your concern lie in three parts.
>
> 1) The first concern is our contribution to the field, for which we provide a comparison to the SOTA related works of SNN robustness (see the section of **TO All Reviewers**) and add the literature on SNN robustness in the revised paper.
> 2) The second concern is the difference from the previous work, for which we explain that the biggest differences lie in the theoretical derivations for spiking Lipschitz analysis. We would like to note that the calculation scheme of Lipschitz constant of ANNs cannot be directly applied to SNNs with discrete activation and time-series processing capability, which forms a difficulty in determining Lipschitz constant of SNNs.
> 3) The third concern is about the ablation study, for which we clarify the meaning of MIX & REG and the result of our ablation study.
>
> Given these facts and positive feedbacks from other reviewers, we sincerely hope that our feedback could settle and answer your concerns. We hope you could reconsider and improve your initial rating. Also, if you have any further questions or comments, please let us know, and we are glad to give further responses.

---

### Official Review · Reviewer_UVTJ · 2022-07-12

**Rating:** 5
**Confidence:** 5
**Soundness:** 2 fair
**Presentation:** 2 fair
**Contribution:** 2 fair

**Summary:**

The paper proposes robust regularization during SNN training to motivate and inspire improved robustness for the trained models.

**Questions:**

Please see weakness.

**Ethics Review Area:**

["I don’t know"]

**Limitations:**

N/A.

**Strengths And Weaknesses:**

## Weakness

### Technical

1. The authors missed a very important point, as earlier literature has already showed the effectiveness of robustness when the input is rate coded [1] and significantly less robustness when input is direct coded [2]. This key aspect is completely overlooked in the manuscript.

2. The authors made a strong claim in L208, without any empirical validation of the fact whether it happens or not. Particularly, when the model is further fine tuned in SNN domain. Hence, the motivation of the work is weak.

3. The paper does not clearly mention what might be their inspiration to use BN in their SNN models as the model definitions are never explicitly mentioned.

4. During training the authors used FGSM attack variants, and during testing they got robustness against PGD variants. This raises significant question about the experimental set up.

5. The authors should proof the robustness is real as this might easily due to gradient obfuscation, and to me it already fails the gradient obfuscation.

6. Authors mentioned they mixed and matched different FGSM attacked images for training to make the attack during training more diverse, no proof (theoretical or empirical) is provided.

7. Analysis of additional training cost is missing. [2] provided improvement in robustness without any additional training cost.

Overall the paper lacks motivation and enough contribution. Comparison with SOTA is also missing. I would encourage the authors to work on the said issues to make the paper better.

[1] Inherent Adversarial Robustness of Deep Spiking Neural Networks: Effects of Discrete Input Encoding and Non-Linear Activations, ECCV 2020.

[2] HIRE-SNN: Harnessing the Inherent Robustness of Energy-Efficient Deep Spiking Neural Networks by Training With Crafted Input Noise, ICCV 2021.

---

> ### Author Response · Authors · 2022-08-02
> **Response to Reviewer UVTJ (Part 1/4)**
>
> Thank you for your detailed and insightful comments. We would like to address your concerns and answer your questions here.
>
> ## 1. The authors missed a very important point, as earlier literature has already showed the effectiveness of robustness when the input is rate coded [1] and significantly less robustness when input is direct coded [2]. This key aspect is completely overlooked in the manuscript.
>
> Thanks for your suggestion. [2] considered rate coding and proposed to train SNNs with crafted-noise under the observation that SNNs are less robust using direct coding. This also inspires us to propose a robust learning scheme. We have revised the paper according to your suggestion and adjusted the motivation and related work. Our work is actually based on the insights of [2] and direct coding, which is complementary to the work of [1][2]. We focus on discussing the robustness of inter-layer spike communication using the theoretical derivation of the spiking Lipschitz coefficient.
>
> Revised content:
>
> *The operation mechanism and structure of SNN are similar to those of the biological brain, and studying its response to perturbation can help us understand how the human brain works. SNN is recognized as a new potential candidate with adversarial robustness due to its input coding and neuronal dynamics \citep{perez2021neural, leontev2021robustness}. **Among the coding schemes commonly used in SNN, constant input coding is considered to be more susceptible to disturbances than others, like Poisson coding~\citep{sharmin2020inherent}.** Therefore, \cite{kundu2021hire} suggested that careful training is required for constant input coding. In this setting, SNNs are now facing more challenges than typical ANNs. Because the key to constructing SNN gradient-based attacks is back-propagation, which is the same as that of ANNs. However, compared with ANNs, SNNs can learn through various gradient approximations. Therefore, combining various differentiable approximations and attack methods will pose a more severe threat to SNN~\citep{liang2021exploring}.*
>
> *The inter-layer communication of SNN is through spikes with a time dimension, which is very different from ANN. Therefore, one question can be raised naturally: whether and to what extent does spike communication detain adversarial invulnerability? And, are there training tools that can help SNNs defend against the threats described above? This paper aims to extend the Lipschitz analysis theory to spike representation and propose a more robust training algorithm on this basis.*
>
> ## 2. The authors made a strong claim in L208, without any empirical validation of the fact whether it happens or not. Particularly, when the model is further fine tuned in SNN domain. Hence, the motivation of the work is weak.
>
> Thanks for your suggestion. We would like to claim that our intention of using spike train distance is to give a solution of distance with the intention of reducing the loss of temporal information encoded in the spike trains. Hence, we adopt the definition of the spike distance function in Eq.12, which contains both rate and temporal information. Our inspiration is from the field of neuroscience, where various spike train distances are proposed and applied.
> We reorganize the statement in Section 4.1 as follows:
>
> *\cite{kundu2021hire} used this rate-based distance to bridge the robustness of ANN and SNN. Spike trains in SNN not only contain rate information but also have a temporal  structure. To evaluate the distance in the spike train space, various kernel methods are proposed for neuronal identification and encoding~\citep{weng2018towards}. Inspired by these works, we propose to model the distortion of the spike response using the spike train distance, which can bridge the robustness of SNN to the discovery of neuroscience and is also sensitive to the change of both firing rate and temporal information.*
>
> ## 3. The paper does not clearly mention what might be their inspiration to use BN in their SNN models as the model definitions are never explicitly mentioned.
>
> Thanks for your suggestion. We would like to clarify that Batch Normalization has been used in SNN direct training with BPTT [3][4], which plays an important role in mitigating gradient explosion and vanishing. Recent works explore various BN variants for SNN direct training [5][6]. The BN variant proposed in [6] is considered to play a role in the adversarial robustness of SNNs. Therefore, we consider using BN in this paper. We have clarified the motivation for using BN in the revised paper.

---

> > ### Author Response · Authors · 2022-08-02
> > **Response to Reviewer UVTJ (Part 2/4)**
> >
> > ## 4. During training the authors used FGSM attack variants, and during testing they got robustness against PGD variants. This raises significant question about the experimental set up.
> >
> > We would like to note that our experimental setup is similar to the works in adversarial robustness, which train with some attack methods and evaluate models with the same or more powerful adversarial attacks [7][8][9][10]. The setting of FGSM-training in our paper is the same as a subfield called fast adversarial training. Here we give two examples to clarify it. [7] proposed that using a single-step attack during training is an efficient and feasible scheme to improve model robustness. Random initialized FGSM-training has similar effects as PGD-training, but with lower computational cost. Similarly, the role of FGSM-training in adversarial learning is also recognized by [8]. Generally, [7] and [8] have experimental settings of single-step attack training and multi-step attack testing.
> >
> > ## 5. The authors should proof the robustness is real as this might easily due to gradient obfuscation, and to me it already fails the gradient obfuscation.
> >
> > Thanks for your suggestion. In order to identify the attack effectiveness of differentiable approximations of CBA, BPTT, BPTR for SNN, we adopt the checklist mentioned in [2] to systematically analyze the gradient obfuscation of the three schemes. The analysis is mainly based on Table 1 and Table 2 in the main text. Our brief results are presented in Table R2. Since CBA has failed in Test (1), subsequent tests on CBA are not considered. CBA is also not used in our robust training scheme. Detailed analyses are presented in the appendix.
> >
> > **Table R2: Checklist for characteristic behaviors caused by obfuscated and masked gradients.**
> > | Items to identify gradient obfuscation | CBA | BPTR | BPTT |
> > | -------- | -------- |-------- |-------- |
> > | (1) Single-step attack performs better compared to iterative attacks | Fail | Pass | Pass |
> > | (2) Black-box attacks performs better compared to white-box attacks | NA | Pass | Pass |
> > | (3) Increasing perturbation bound can’t increase attack strength | NA | Pass | Pass |
> > | (4) Unbounded attacks can’t reach ∼100% success | NA | Pass | Pass |
> > | (5) Adversarial example can be found through random sampling | NA | Pass | Pass |
> >
> > *We design and summarize three differentiable approximations, i.e. CBA, BPTT, BPTR, which can be deployed in gradient-based attacks to show the vulnerability of SNNs. The main concern of the gradient obfuscation lies in the inaccurate of updating gradients. In particular, the performance of the three differentiable approximations was checked against the five tests that can identify gradient obfuscation as done in \cite{kundu2021hire}. Our analysis is mainly based on the quantification results in Table 1 and Table 2 in the main text. Also, this will explain the reason why we choose BPTT and BPTR in the procedure of the mixed training.*
> >
> > *As shown in Tab. 1, for all the trials, the performance of single-step FGSM is worse than its iterative counterpart PGD except for that of the WRN-16 experiment for CIFAR-100 (Attacked Accuracy: FGSM 37.68\% v.s. PGD 43.87\%). Thus, the CBA approximation has the potential not to provide powerful enough attacks.*
> >
> > *Hence, the rest of the analysis is about BPTT and BPTR. The results in Tables 1 and 2 certify the success of BPTT and BPTR approximation in terms of Test(1) in Table R2. To verify Test(2), we conduct black-box attacks on the proposed models and the vanilla ones. The black-box perturbation performs weaker in Table 2, and Test(2) is satisfied. To verify Test(3)(4), we analyze VGG-11 on CIFAR10 with increasing attack bound. In Figure A1, the classification accuracy decreases as we increase $\epsilon$ and finally reach an accuracy of random guessing. As suggested in [2], Test(5) "can fail only if gradient-based attacks cannot provide adversarial examples for the model to misclassify". To sum up, we found no gradient obfuscation for the BPTT and BPTR approximation, which are suitable for adversarial training and testing.*

---

> > > ### Author Response · Authors · 2022-08-02
> > > **Response to Reviewer UVTJ (Part 3/4)**
> > >
> > > ## 6. Authors mentioned they mixed and matched different FGSM attacked images for training to make the attack during training more diverse, no proof (theoretical or empirical) is provided.
> > >
> > > Thanks for your valuable suggestion. We would like to note that the reason we mix the different FGSM variants are to find generalization on multiple adversarial methods, which mainly consists of three folds:
> > >
> > > (1) To improve the effect on PGD attacks. The motivation is based on some conclusions from previous works. [7] suggested that random initialized FGSM-training has similar effects as PGD-training, but with lower computational cost. [11] focused on the robustness of quantized networks and claimed that R-FGSM introduced randomness, making it less likely to cause gradient masking than FGSM. In fact, in the experiments of [11], R+FGSM training is just as effective as PGD adversarial training. Also, [12] showed that adversarial robustness against PGD attacks could be achieved with RFGSM-based training.
> > >
> > > (2) To improve the generalization of the noises. The adversarial distortion produced by BPTT and BPTR are of different nature as the formation is different (compare Eq.9 for BPTT and Eq.11 for BPTR). Besides, the mixed attack can bring about a noisy variant of FGSM. As [13] suggested, noisy FGSM moves the perturbation boundary of the FGSM attack, which leads to a larger variety of adversarial examples.
> > >
> > > (3) To improve the overall computational efficiency. BPTR is proven to be an efficient and powerful adversarial attack. The efficiency is proven by the analysis of overall testing time. BPTR is about 3x faster than BPTT in testing (Please refer to Analysis of Computational Cost in the revised appendix). In addition, BPTR has passed the gradient obfuscation test (Please refer to Analysis of Gradient Obfuscation in the revised appendix). Overall, the addition of BPTR can shorten the time consumed by backward pass without reducing the effectiveness.
> > >
> > > ## 7. Analysis of additional training cost is missing. [2] provided improvement in robustness without any additional training cost.
> > >
> > > Thank you for pointing this out. Our proposed training scheme included three differential approximations. For robust training, each step of the update process mainly consists of two forward and two backward passes. For testing, each step of the update has two forward passes and one backward pass.
> > >
> > > We evaluate the time of the testing process of three approximations (CBA, BPTR, BPTT) instead. During the tests, we fix the mini-batch size to 64 and run the test on a NVIDIA 3090 GPU. The results are presented in the revised paper (Please refer to the Appendix). It turns out that the time consumption ratio of the three methods (CBA:BPTR: BPTT) is about 1:1:3. Please refer to the appendix of the revised paper.
> > >
> > > We believe the time consuming cost by proposed training scheme will benefit from the addition of BPTR-based adversarial training, compared with the vanilla pure BPTT attack and training.

---

> > > > ### Author Response · Authors · 2022-08-02
> > > > **Response to Reviewer UVTJ (Part 4/4)**
> > > >
> > > > [1] Sharmin, S., Rathi, N., Panda, P., & Roy, K. (2020). Inherent adversarial robustness of deep spiking neural networks: Effects of discrete input encoding and non-linear activations. In European Conference on Computer Vision, 399-414.
> > > >
> > > > [2] Kundu, S., Pedram, M., & Beerel, P. A. (2021). Hire-snn: Harnessing the inherent robustness of energy-efficient deep spiking neural networks by training with crafted input noise. In Proceedings of the IEEE/CVF International Conference on Computer Vision, 5209-5218.
> > > >
> > > > [3] Fang, W., Yu, Z., Chen, Y., Huang, T., Masquelier, T., & Tian, Y. (2021). Deep residual learning in spiking neural networks. Advances in Neural Information Processing Systems, 34, 21056-21069.
> > > >
> > > > [4] Li, Y., Guo, Y., Zhang, S., Deng, S., Hai, Y., & Gu, S. (2021). Differentiable spike: Rethinking gradient-descent for training spiking neural networks. Advances in Neural Information Processing Systems, 23426-23439.
> > > >
> > > > [5] Zheng, H., Wu, Y., Deng, L., Hu, Y., & Li, G. (2021). Going deeper with directly-trained larger spiking neural networks. In Proceedings of the AAAI Conference on Artificial Intelligence, 11062-11070.
> > > >
> > > > [6] Kim, Y., & Panda, P. (2020). Revisiting batch normalization for training low-latency deep spiking neural networks from scratch. Frontiers in neuroscience, 1638.
> > > >
> > > > [7] Wong, E., Rice, L., & Kolter, J. Z. (2019). Fast is better than free: Revisiting adversarial training. In International Conference on Learning Representations.
> > > >
> > > > [8] Andriushchenko, M., & Flammarion, N. (2020). Understanding and improving fast adversarial training. Advances in Neural Information Processing Systems, 33, 16048-16059.
> > > >
> > > > [9] Shafahi, A., Najibi, M., Ghiasi, M. A., Xu, Z., Dickerson, J., Studer, C., ... & Goldstein, T. (2019). Adversarial training for free!. Advances in Neural Information Processing Systems, 32.
> > > >
> > > > [10] Ortiz-Jiménez, G., Modas, A., Moosavi-Dezfooli, S. M., & Frossard, P. (2021). Optimism in the face of adversity: Understanding and improving deep learning through adversarial robustness. Proceedings of the IEEE, 109(5), 635-659.
> > > >
> > > > [11] Lin, J., Gan, C., & Han, S. (2018). Defensive Quantization: When Efficiency Meets Robustness. In International Conference on Learning Representations.
> > > >
> > > > [12] Xu, Z., Shafahi, A., & Goldstein, T. (2020). Exploring Model Robustness with Adaptive Networks and Improved Adversarial Training.
> > > >
> > > > [13] Schwinn, L., Raab, R., & Eskofier, B. (2020). Towards rapid and robust adversarial training with one-step attacks. arXiv preprint arXiv:2002.10097.

---

> > > > > ### Comment · Reviewer_UVTJ · 2022-08-02
> > > > > **Further clarification needed**
> > > > >
> > > > > I thank the authors for the detailed rebuttal, and few of my concerns are well addressed. However, I have the following concerns remaining:
> > > > >
> > > > > 1. It is well researched that using BNs are apparently [1] not good to get adversarial robustness. So, I would strongly encourage the authors to provide results on models without BNs particularly when providing results on robustness.
> > > > > There are other alternate approaches that can handle the "lack of BN issues", please refer to [2]. So, a discussion on this is necessary.
> > > > >
> > > > > 2. I understand that randomly initialized FGSM provides better results [3], however, it is not clear whether the FGSM training of the current manuscript follows this.
> > > > >
> > > > > 3. The authors should further change the claim of L208, as they dont have any empirical or theoretical evidence to : "which **may bridge** the robustness of SNN to the discovery of neuroscience and is also sensitive to the change of both firing rate and temporal information."
> > > > >
> > > > > 4. To my understanding of the rebuttal, the proposed method has additional training cost, and thus is not fair to directly compare with [4,5]. It is important for the author to tone down on their claims and show a fair table and detail this in the paper. It also hints at additional inference cost, thus please clearly mention this if I am correct. Otherwise it might pose an incorrect information to the community, as both [4] and [5] tried to show inherent robustness. Having said these, I believe the current manuscript has value, and I will assert my final decision based on that.
> > > > >
> > > > > **Post rebuttal initial rating**: I now increase my score to 5.
> > > > >
> > > > > [1] Removing Batch Normalization Boosts Adversarial Training, ICML 2022.
> > > > >
> > > > > [2] Spike-thrift: Towards Energy-Efficient Deep Spiking Neural Networks by Limiting Spiking Activity via Attention-Guided Compression, WACV 2021.
> > > > >
> > > > > [3] Fast is better than free: Revisiting adversarial training, ICLR 2020.
> > > > >
> > > > > [4] Inherent adversarial robustness of deep spiking neural networks: Effects of discrete input encoding and non-linear activations. In European Conference on Computer Vision, 2020.
> > > > >
> > > > > [5] Hire-snn: Harnessing the inherent robustness of energy-efficient deep spiking neural networks by training with crafted input noise. In Proceedings of the IEEE/CVF International Conference on Computer Vision, 2021.

---

> > > > > > ### Author Response · Authors · 2022-08-07
> > > > > > **Further Response to Reviewer UVTJ (Part 1/2)**
> > > > > >
> > > > > > Thank you for your continued interest and constructive comments on our work. We are
> > > > > > to know that we have resolved some of your concerns. We would like to answer your remaining questions in the following.
> > > > > >
> > > > > > ## 1. It is well researched that using BNs are apparently [1] not good to get adversarial robustness. So, I would strongly encourage the authors to provide results on models without BNs particularly when providing results on robustness. There are other alternate approaches that can handle the "lack of BN issues", please refer to [2]. So, a discussion on this is necessary.
> > > > > >
> > > > > > **Tabel R3: Effect of Batch Normalization**
> > > > > > |                  | RAT (with BN) | Vanilla (with BN) | RAT (w/o BN)  | Vanilla (w/o BN) |
> > > > > > | ---------------- | ------------- | ----------------- | ------------- |:---------------- |
> > > > > > | CLEAN            | 82.03         | 90.170            | 75.820        | 88.880           |
> > > > > > | FGSM(BPTR/BPTT)  | 38.890/33.410 | 8.400/6.140       | 33.440/27.530 | 15.110/9.000     |
> > > > > > | RFGSM(BPTR/BPTT) | 58.060/53.460 | 25.040/17.880     | 52.950/48.890 | 33.300/19.650    |
> > > > > > | PGD(BPTR/BPTT)   | 28.390/16.530 | 0.280/0.030       | 27.380/19.460 | 1.100/0.040      |
> > > > > >
> > > > > > Thanks for pointing it out. We agree that BN is recognized to have a negative impact on model robustness in some literature. Hence, we have done further research on the effect of Batch Normalization on our proposed RAT scheme. We trained the spiking version of VGG5 on the CIFAR-10 dataset with four different settings: RAT+BN, Vanilla+BN, RAT without BN, and Vanilla without BN. The results are presented in Table R3.
> > > > > >
> > > > > > From the table, we find that for vanilla models without proposed RAT, the absence of BN helps improve the robustness. Training with RAT has promoted the robustness of models, either with or without BN. The robustness of RAT improves with BN, as BN can increase the clean accuracy. Therefore, according to Table R3, we can roughly think in terms of robustness that: RAT (with BN) ) > RAT (w/o BN) > Vanilla (w/o BN) > Vanilla (with BN).
> > > > > >
> > > > > > As for SNN robustness without BN, we have added a discussion in the section of 'Conclusions and Discussions', where we pointed out the problem with BN and alternate approaches to train SNN without BN. We are glad to investigate it further in future work.
> > > > > >
> > > > > > Added Contents in Conclusions and Discussions:
> > > > > > *Besides, recent works have shown that SNN can achieve good results without BN~\cite{kundu2021spike}[2]. Note that BN are included in our model, which may be harmful to the robustness~\citep{wang2022removing}[1]. Thus valuable future research directions will include how to train robust SNNs while getting rid of the adverse effects of BN.*
> > > > > >
> > > > > > ## 2. I understand that randomly initialized FGSM provides better results [3], however, it is not clear whether the FGSM training of the current manuscript follows this.
> > > > > >
> > > > > > Yes, we would like to note that we follow the dedicated works of [3], [4], [5] mainly for the reason that the FGSM and RFGSM training has become a powerful baseline method. We are currently engaged in empirical proof of FGSM training. Due to the recent limitations of our computing resources, our experiments are still ongoing. We would like to provide the results in the final version.
> > > > > >
> > > > > > ## 3. The authors should further change the claim of L208, as they dont have any empirical or theoretical evidence to : "which may bridge the robustness of SNN to the discovery of neuroscience and is also sensitive to the change of both firing rate and temporal information."
> > > > > >
> > > > > > Thanks for your suggestion. We have rewritten the sentence according to your advice in the revised manuscript.

---

> > > > > > > ### Author Response · Authors · 2022-08-07
> > > > > > > **Further Response to Reviewer UVTJ (Part 2/2)**
> > > > > > >
> > > > > > > ## 4. To my understanding of the rebuttal, the proposed method has additional training cost, and thus is not fair to directly compare with [4,5]. It is important for the author to tone down on their claims and show a fair table and detail this in the paper. It also hints at additional inference cost, thus please clearly mention this if I am correct. Otherwise it might pose an incorrect information to the community, as both [4] and [5] tried to show inherent robustness.
> > > > > > >
> > > > > > > Thanks for the feedbacks and constructive suggestions. We agree that the improvement of robustness of our proposed training scheme is at a cost. We would like to note that the cost is actually not that high. Compared to the setting of typical adversarial training that only uses BPTT differentiable approximation, we mix fast and effective BPTR to alleviate the drop in training efficiency. At the same time, we update the weights of the orthogonal regularization by sampling instead of updating all weights, as suggested in [6]. This can also help reduce the computational overhead.
> > > > > > >
> > > > > > > To give a fair comparison of the related works, we have updated Table R1 in 'To All Reviewers' to show the additional training cost. In addition, we have added a discussion on additional training costs, which is temporally put in the appendix. We will incorporate your suggestion and move the discussion of training cost in our final version where one additional page allows us to present the comparison of the SOTA works along with the discussion of the training cost.
> > > > > > >
> > > > > > > Additional discussion:
> > > > > > > *It is worth noting that although our training algorithm improves the robustness of SNNs, it comes at a cost compared to the work of Sharmin et al.[4] and Kundu et al.[5]. The cost is mainly reflected in the training time. First, our training includes time to generate adversarial noise. Adversarial learning is a common scheme to improve robustness, and generating adversarial examples using only BPTT differentiable approximation in SNN is a time-consuming operation. Our algorithm mitigates the increase in training time by mixing in a faster yet efficient BPTR approximation. In addition, the orthogonal regularization of the weights is computed every update, which also increases the training time. Solutions to reduce the time consumption of regularization include sampling fewer weights for regularizing, or reducing the number of regularization updates.*
> > > > > > >
> > > > > > > [1] Wang, H., Zhang, A., Zheng, S., Shi, X., Li, M., & Wang, Z. (2022, June). Removing Batch Normalization Boosts Adversarial Training. In International Conference on Machine Learning, 23433-23445.
> > > > > > >
> > > > > > > [2] Kundu, S., Datta, G., Pedram, M., & Beerel, P. A. (2021). Spike-thrift: Towards energy-efficient deep spiking neural networks by limiting spiking activity via attention-guided compression. In Proceedings of the IEEE/CVF Winter Conference on Applications of Computer Vision, 3953-3962.
> > > > > > >
> > > > > > > [3] Wong, E., Rice, L., & Kolter, J. Z. (2019, September). Fast is better than free: Revisiting adversarial training. In International Conference on Learning Representations.
> > > > > > >
> > > > > > > [4] Andriushchenko, M., & Flammarion, N. (2020). Understanding and improving fast adversarial training. Advances in Neural Information Processing Systems, 33, 16048-16059.
> > > > > > >
> > > > > > > [5] Shafahi, A., Najibi, M., Ghiasi, M. A., Xu, Z., Dickerson, J., Studer, C., ... & Goldstein, T. (2019). Adversarial training for free!. Advances in Neural Information Processing Systems, 32.
> > > > > > >
> > > > > > > [6] Cisse, M., Bojanowski, P., Grave, E., Dauphin, Y., & Usunier, N. (2017, July). Parseval networks: Improving robustness to adversarial examples. In International Conference on Machine Learning, 854-863.
> > > > > > >
> > > > > > > [4] Sharmin, S., Rathi, N., Panda, P., & Roy, K. (2020, August). Inherent adversarial robustness of deep spiking neural networks: Effects of discrete input encoding and non-linear activations. In European Conference on Computer Vision, 399-414.
> > > > > > >
> > > > > > > [5] Kundu, S., Pedram, M., & Beerel, P. A. (2021). Hire-snn: Harnessing the inherent robustness of energy-efficient deep spiking neural networks by training with crafted input noise. In Proceedings of the IEEE/CVF International Conference on Computer Vision, 5209-5218.

---

> > > > > > > > ### Comment · Reviewer_UVTJ · 2022-08-07
> > > > > > > > **Thanks for the additional experiments**
> > > > > > > >
> > > > > > > > Dear authors,
> > > > > > > >
> > > > > > > > I have read the recent response and have seen the results. I want to make one follow up point:
> > > > > > > >
> > > > > > > > 1. It would be good if you can have insight on why with your approach the with BN model is performing better.
> > > > > > > >
> > > > > > > > Also, please update the results (that are currently running), on the rebuttal response.
> > > > > > > >
> > > > > > > > The authors have put a significant effort in rectifying their lacking and I am pleased with the additional set of experiments and clarifications. Given the authors **will include experiments, reasoning for all the concerns raised by me**, I tend to accept this paper now.

---

> > > > > > > > > ### Author Response · Authors · 2022-08-09
> > > > > > > > > **Some insights on Batch Norm**
> > > > > > > > >
> > > > > > > > > **Tabel R4: Layerwise Matrix Norm of Batch Normalization**
> > > > > > > > > | Performance      | RAT (w/ BN)       | RAT (w/o BN)     | Vanilla (w/o BN)     | Vanilla (w/ BN)       |
> > > > > > > > > | ---------------- | ----------------- | ---------------- |:-------------------- | --------------------- |
> > > > > > > > > | FGSM (BPTR/BPTT) | 38.890/33.410     | 33.440/27.530    | 15.110/9.000         | 8.400/6.140           |
> > > > > > > > > | **Matrix Norm**  | **RAT (with BN)** | **RAT (w/o BN)** | **Vanilla (w/o BN)** | **Vanilla (with BN)** |
> > > > > > > > > | Layer 1          | 1.10              | 1.34             | 5.44                 | 6.27                  |
> > > > > > > > > | Layer 2          | 1.07              | 1.21             | 3.80                 | 3.25                  |
> > > > > > > > > | Layer 3          | 1.11              | 1.22             | 3.35                 | 3.44                  |
> > > > > > > > > | Layer 4          | 1.96              | 2.09             | 3.06                 | 4.10                  |
> > > > > > > > > | Layer 5          | 2.18              | 2.21             | 4.71                 | 4.81                  |
> > > > > > > > > | Layer 6          | 1.43              | 1.39             | 4.36                 | 4.92                  |
> > > > > > > > >
> > > > > > > > > Thanks for your acknowledgment of our efforts. For batch normalization, we would like to present some numeric results and insights to better understand the impact of batch normalization in the context of Lipschitz analysis. Table R4 reports the matrix norms of the weights of each layer in spiking VGG5. Here we rearrange the columns so that the columns are in descending order of robustness. We observe a phenomenon where the matrix norm of the weights at each layer increases as the robustness of the model decreases under different experimental settings, which implies that there is a correlation between the robustness of the SNN and the matrix norm of the weight (the Lipschitz constant of the linear layer).
> > > > > > > > >
> > > > > > > > > We can try to explain the results in terms of the magnitude of the Lipschitz constant and robustness. Without using the RAT scheme, the network without BN has a similar but smaller layerwise Lipschitz constant and, therefore, more robust. When using the RAT scheme, BN can normalize the input of each layer, making regularization training in RAT more effective, resulting in a smaller Lipschitz constant and better robustness. In addition, for the direct training of deep SNN, BN can improve the clean accuracy to a certain extent.
> > > > > > > > >
> > > > > > > > > Thus, from the perspective of Lipschitz constraint, the technique of BN is also necessary. We hope our explanations will help you understand and support our work.

---

### Author Response · Authors · 2022-08-02
**To All Reviewers**

We are grateful to all the reviewers for their insightful feedback. We would like to address the common concerns about the comparison with the state-of-the-art in this general response.

## Comparison with the SOTA

We compare our methods with the state-of-the-art models and report the results in Table R1. As SNN adversarial robustness is a very new research field and has not been researched thoroughly, here we compare two SOTA works [1][2] that highly relate to our work. The evaluation is based on the VGG-11 on the CIFAR-100 dataset. The noise budget has been fixed to $\epsilon=8/255$ for FGSM and $\alpha=0.01,step=7$ for PGD, and the attack is based on the surrogate gradient produced by BPTT.

In Tabel R1, one can find that our training scheme outperforms the others in terms of both clean accuracy and perturbed accuracy. The performance of accuracy attacked by FGSM is 25.86% for our work, higher than that of Sharmin et al.[1] (15.5%) and Kundu et al.[2] (22.0%). Moreover, our clean accuracy (70.89%) is higher than that of Sharmin et al.[1] (64.4%) and Kundu et al.[2] (65.1%). This implies that our proposed method can bring better generalization compared to other SOTA robust models.

We have added the comparison in the revised paper (please refer to the appendix). In addition, to show a fair comparison of the synthetical performance, we have also included a row to present the additional training cost. For a detailed discussion on training cost, please refer to the appendix.

**Table R1: Performance comparison with the SOTA models.**
| Attack by BPTT                    | Proposed training | Sharmin et al. [1] | Kundu et al. [2] | Regular BPTT training |
| --------------------------------- | ----------------- | ------------------ | ---------------- | ------------- |
| FGSM                              | 25.86             | 15.5               | 22.0             | 5.30          |
| PGD                               | 10.38             | 6.3                | 7.5              | 0.02          |
| Clean                             | 70.89             | 64.4               | 65.1             | 73.33         |
| Additional Training Cost | Regularized Training | - | - | - |

[1] Sharmin, S., Rathi, N., Panda, P., & Roy, K. (2020). Inherent adversarial robustness of deep spiking neural networks: Effects of discrete input encoding and non-linear activations. In European Conference on Computer Vision, 399-414.

[2] Kundu, S., Pedram, M., & Beerel, P. A. (2021). Hire-snn: Harnessing the inherent robustness of energy-efficient deep spiking neural networks by training with crafted input noise. In Proceedings of the IEEE/CVF International Conference on Computer Vision, 5209-5218.

---

### Comment · Reviewer_UVTJ · 2022-08-07
**Post rebuttal acknowledgement**

I have updated my score to borderline accept (conditional), primarily due to the authors' apparently thorough rebuttal. However, I believe this paper requires further input from authors and necessary discussion among reviewers for consensus.

---

### Meta-Review · Area_Chair_oE4T · 2022-08-30

**Recommendation:** Accept
**Confidence:** Certain

**Metareview:**

This paper proposes an adversarial training method for Spike neural networks. One challenge is that spike networks are non-differentiable and the paper develops various gradient approximation methods and builds on previous attack methods like FGSM and PGD with approximate gradients. An additional innovation is the development of a regularization method that estimates Lipschitz constants. Estimating Lipschitz constants of spike neural networks is another technical challenge and the paper develops a rigorous bound using a concept they call spike distance. which is an upper bound to the normal Lipschitz contanst.

Several concerns were raised by the reviewers including incomplete discussion of prior work, clarifications on the performed ablation study and comparison to prior SOTA. Overall the authors did a good job in their rebuttal and discussion to convince the revewers and this meta-reviewer that the paper merits publication. This is a somewhat niche problem setting but the paper has several theoretical and practical innovations that are interesting and suitable for publication.


**Award:**

No

---

### Decision · Program_Chairs · 2022-09-14

Accept